# Comparative analysis of the effects of cyclophosphamide and dexamethasone on intestinal immunity and microbiota in delayed hypersensitivity mice

Xiangling Li[1‡], Ruyan Wen[1‡], Ben Chen[1‡], Xia Luo[2], Lu Li[1], Jun Ai[3], Junlong Yu[3]*

**1** Guangxi Key Laboratory of Chinese Medicine Foundation Research, Guangxi University of Chinese Medicine, Nanning, Guangxi, China, **2** School of Pharmaceutical Sciences, Guangzhou University of Chinese Medicine, Guangzhou, China, **3** School of Basic Medicine, Guangxi University of Chinese Medicine, Nanning, Guangxi, China

‡ XL, RW and BC contributed equally to this work and share first authorship.
* haoyunlinglong@163.com

## Abstract

### Background

The T cell-mediated delayed-type hypersensitivity (DTH) response is critical for elucidating cellular immune mechanisms, especially the role of memory T cells upon antigen re-expo-sure. This study aimed to investigate the specific effects of the immunosuppressive drugs Cyclophosphamide (CY) and Dexamethasone (DEX) on intestinal immunity and microbiota in a DTH mouse model, contributing to a more nuanced understanding of their immunomod-ulatory mechanisms.

### Methods

Female BALB/c mice were sensitized to 2,4-dinitrofluorobenzene (DNFB) and randomly allocated into control, CY, and DEX groups. The impact of CY and DEX on immune function was assessed through measurement of thymus and spleen indices, lymphocyte proliferation in mesenteric lymph nodes (MLNs) using MTT assay, and flow cytometric analysis of T cell subsets and TCR expression. Intestinal secretory IgA (sIgA) was quantified by ELISA, and gut microbiota diversity was evaluated using 16S rRNA gene sequencing.

### Results

CY and DEX significantly reduced the immune function in DNFB-induced sensitized mice, as indicated by decreased thymus and spleen indices, MLN enlargement, intestinal sIgA content, and ear swelling degree. Flow cytometry revealed that CY increased the proportion of total CD3+ T cells but reduced CD3+CD69+ activated T cells and CD3+TCRγ/δ+ T cells, while DEX increased CD3+CD4+ helper T cells. Both drugs induced distinct changes in gut microbiota diversity and structure, with CY enhancing α diversity and DEX reducing it.

**Data Availability Statement:** All relevant data are within the manuscript and its Supporting information files.

**Funding:** This study was supported by Guangxi University of Chinese Medicine Doctoral Program (2020BS011 and 2021BS019), Guangxi Key Laboratory of Chinese Medicine Foundation Research Projects (22-065-53-02), Projects to improve the basic scientific research capacity of young and middle-aged people (2020KY07002) and Guangxi first-class discipline Clinical basis of Traditional Chinese Medicine (TCM) (0502300304). The funders had no role in study design, data collection and analysis, decision to publish, or preparation of the manuscript.

**Competing interests:** The authors have declared that no competing interests exist.

## Conclusions

The study demonstrates that CY and DEX have distinct regulatory effects on the immune organ index, distribution of T cell subsets, and diversity and structure of gut microbiota on DTH-induced immune responses mice, suggesting their differential influence on intestinal mucosal immunity. These findings have implications for the development of targeted immunotherapies and understanding the interplay between immunosuppressive drugs and gut microbiota.

## Introduction

Delayed-type hypersensitivity (DTH), a form of T cell-mediated immune response, is pivotal for understanding the intricacies of cellular immunity, particularly the roles of memory T cells in antigen re-encounter scenarios [1, 2]. The DTH reaction, characterized by local inflammation post-secondary exposure to an antigen, serves as a critical model for deciphering the pathophysiological mechanisms underlying T cell-mediated inflammatory processes [3].

Immunosuppressive agents, pivotal in treating autoimmune and inflammatory disorders and preventing allograft rejection, include Cyclophosphamide (CY) and Dexamethasone (DEX) [4, 5]. CY, an alkylating agent, and DEX, a potent glucocorticoid, are distinguished by their distinct yet overlapping mechanisms of action [6, 7]. While their systemic immunosuppressive effects are well-documented, their specific influences on intestinal immunity and microbiota in the context of DTH have not been comprehensively explored.

The gastrointestinal tract, with its diverse microbiota, plays a significant role in immune homeostasis and has become a recognized target for the immunomodulatory actions of immunosuppressive drugs [8, 9]. Disruptions in the gut microbiota due to these agents can precipitate dysbiosis, with implications for systemic immune responses and inflammation [10, 11]. Both CY and DEX can successfully induce immunosuppression in various models [12, 13], and a dose of 80 mg/kg CY used for 5 consecutive days can successfully induce immunodeficiency in mice [14]. Based on our previous time (24h, 48h, 72h) and dosage (DEX: 200mg/kg, 120mg/kg, 40mg/kg, 24mg/kg or Cy: 300mg/kg, 200mg/kg, 100mg/kg, 50mg/kg) studies of Cy and DEX [15], the current study aims to dissect the comparative impact of a single high dose of CY (100 mg/kg) and DEX (40 mg/kg) on intestinal immunity and microbiota in a DTH mouse model.

The objective of this research is to determine the capacity of these agents to induce immunosuppression in both systemic and intestinal compartments, thereby contributing to a more sophisticated understanding of their mechanisms. This study seeks to uncover the nuances of how CY and DEX modulate intestinal mucosal immunity and microbiota, which could have significant implications for therapeutic strategies and the development of targeted immunotherapies.

## Materials and methods

### Animals and ethics

Female BALB/c mice, aged 6 weeks and weighing 20 ± 2 g, were obtained from the Guangdong Medical Experiment Animal Center, Guangzhou, China. Following a one-week acclimation period under specific pathogen-free (SPF) conditions, the mice were group-housed. Each group consisted of 6 mice, and a total of 18 mice were evenly distributed across three groups

corresponding to the Control, CY, and DEX treatments. The group-housing environment was carefully controlled, maintaining a temperature range of 20~26˚C, with a 12 hour light/dark cycle, and providing ad libitum access to food and water. Group housing was chosen to allow for natural social interactions, which are essential for the mice's behavioral and physiological well-being and to reflect more natural conditions for microbiome development.

The housing facility was designed to minimize environmental variations that could affect the gut microbiota, ensuring that all mice within the study were exposed to consistent conditions. Each group was housed in a separate cage to prevent cross-contamination of microbiota between groups while still allowing for social interaction within the group. The study was conducted in strict accordance with the Guide for the Care and Use of Laboratory Animals of the National Institutes of Health, and the protocol was approved by the Laboratory Animal Welfare and Ethics Committee of Guangxi University of Chinese Medicine (Approval No. DW20211205-101). All surgery was performed under sodium pentobarbital anesthesia, and all efforts were made to minimize suffering.

In instances where mice experienced a weight loss exceeding 20%, they were humanely euthanized with pentobarbital sodium at a dosage of 150 mg/kg body weight, administered intraperitoneally. This method ensures the maintenance of high ethical standards throughout the study.

## Sensitization, grouping processing, and sample collection

After a one-week acclimation, hair removal was performed on a 1.5 cm × 1 cm area on the mice's backs. On Day 2, sensitization was initiated by applying 10 μL of 44 mmol 2,4-dinitrofluorobenzene (DNFB) in acetone. The sensitized mice were then divided into three groups: control, CY, and DEX, with six mice per group. On Day 4, the control group received an intraperitoneal injection of 20 mL/kg PBS, the CY group received 100 mg/kg Cyclophosphamide, and the DEX group received 40 mg/kg Dexamethasone.

On Day 6, a challenge was conducted by applying 10 μL of 80 mmol DNFB in acetone to the right ear of each mouse, with the left ear receiving 10 μL of acetone as a control. The following day, Day 7, fecal samples were gathered through direct anal extraction to analysis the gut microbiota, after that, ear tissue samples were collected post-euthanasia using an 8.5 mm punch to measure the degree of ear swelling. The spleen and thymus were removed, weighed, and their organ indices calculated. The mesenteric lymph nodes (MLNs) were extracted to evaluate proliferation and cell subpopulations of lymphocyte. The small intestine was harvested to analysis the intestinal Peyer's Patches (PPs) and determination of Intestinal Secretory IgA. The ear tissue samples were fixed, embedded, sectioned, and stained with hematoxylin and eosin for histological examination.

## Analysis of intestinal Peyer's patches

The small intestine was expeditiously harvested post-euthanasia and rinsed in Phosphate-Buffered Saline (PBS) to visually identify and photograph PPs for quantitative analysis. Segments containing PPs were meticulously excised, cleared of luminal contents with PBS, and fixed in 10% neutral buffered formalin for 24 hours. Paraffin embedding followed, and 4-micron thick sections were prepared. These sections underwent Hematoxylin and Eosin (HE) staining to elucidate the histological structure of PPs, including the assessment of morphological features and the presence of germinal centers indicative of immune activation, as well as the identification of inflammatory cell infiltrates.

## Lymphocyte proliferation from mesenteric lymph nodes (MLNs) was evaluated using a modified MTT assay

MLNs were aseptically extracted, processed through a 200-mesh sieve in cold PBS to yield a single-cell suspension, and washed by centrifugation at $300 \times g$ for 5 minutes. The cell concentration was adjusted to $2 \times 10^6$ cells/mL in RPMI-1640 medium supplemented with 10% FBS, penicillin, and streptomycin.

Cells (200 μL) were seeded in 96-well plates and stimulated with 5 μg/mL Concanavalin A (Con A). The plates were incubated for 68 hours at 37°C in a 5% $CO_2$ atmosphere. Subsequently, 10 μL of MTT solution (5 g/L) was added to each well for an additional 4-hour incubation. The reaction was stopped by centrifugation at $800 \times g$ for 10 minutes to pellet the formazan crystals. The supernatant was removed and 150 μL DMSO was added to each well for solublization. It was then incubated at 37°C for 10 min. After that, the absorbance was read at 570 nm using a multi-function microplate reader (Swiss TECAN), with 630 nm as the reference wavelength.

## Differential flow cytometric assessment of mesenteric lymph node lymphocytes

Mesenteric lymph node (MLN) cells were harvested post-euthanasia, washed in cold PBS, and processed to generate a uniform single-cell suspension. The cells were centrifuged, resuspended, and adjusted to a concentration of $1 \times 10^7$ cells/mL in RPMI-1640 medium supplemented with 10% FBS and antibiotics.

For the flow cytometric analysis, two distinct 100 μL aliquots of the cell suspension were prepared. The first aliquot was stained with a combination of antibodies to delineate T cell populations: PE-Cy7 anti-mouse CD3 (0.25 μg, 25-0031-81), PE anti-mouse CD8 (0.25 μg, MA1-10304), APC anti-mouse CD4 (0.5 μg, 17-0042-82), and FITC anti-mouse CD69 (0.25 μg, 11-0691-82). The second aliquot was stained to evaluate TCR expression: PE-Cy7 anti-mouse CD3 (0.25 μg, 25-0031-81), APC anti-mouse TCR β (0.25 μg, 47-5961-82), and FITC anti-mouse TCR γ/δ (0.25 μg, 11-5711-82). All antibodies were sourced from eBioscience, USA.

Incubation with the antibody cocktails was performed in the dark at 4°C for 30 minutes to prevent photobleaching. Excess antibodies were removed by centrifugation and resuspension in PBS. The samples were then analyzed using a FACScanto Flow Cytometer, and the data were processed with the corresponding software to quantify the proportions of $CD4^+$ and $CD8^+$ T cells, as well as TCR $β^+$ and TCR $γ/δ^+$ cells within the MLNs.

## Determination of Intestinal Secretory IgA in mouse ileum

Secretory IgA (sIgA) in the mouse ileum was quantified by ELISA. A 15 cm section of the ileum was dissected, starting from the appendix, and immediately placed in a Petri dish with 10 mL of cold PBS. The intestinal lumen was flushed with PBS using a gavage needle, and this process was repeated five times to ensure complete collection of luminal contents.

The collected flushings were incubated at 4°C for 30 minutes to allow for the sedimentation of particulate matter, followed by centrifugation at $2000 \times g$ for 5 minutes at 4°C to pellet debris. The supernatant was transferred to 1.5 mL Eppendorf tubes and stored at -20°C until analysis.

Intestinal sIgA levels were measured using a mouse sIgA-specific ELISA kit (E-EL-M1040, elabscience, Wuhan, China) according to the manufacturer's guidelines. The ELISA procedure provided a quantitative assessment of sIgA, which is a key component of the mucosal immune system in the gastrointestinal tract.

## Assessment of gut microbiota abundance and diversity in mice

Gut microbiota analysis was conducted on fecal samples collected on Day 7 by direct anal extraction. Total DNA was extracted from the samples using a commercial DNA extraction kit (Omega Company, USA) following the manufacturer's protocol. The V3-V4 region of the 16S rRNA gene, a standard marker for bacterial diversity studies, was amplified using the universal primer set: 341F (5'-CCTACGGGNGGCWGCAG-3') and 805R (5'-GACTACHVGGGTATCTAATCC-3').

Agarose gel electrophosis at 2% was performed to verify the size of the amplified DNA fragments, ensuring successful amplification of the targeted 16S rRNA gene region. The concentration of the purified amplicons was quantified using a Qubit 3.0 fluorometer, ensuring accurate measurement for subsequent sequencing steps.

The purified and quantified amplicons were subjected to high-throughput sequencing on the Illumina MiSeq™/HiSeq™ platform, which provided comprehensive data on the fecal microbiota composition. The sequencing process generated a detailed profile of operational taxonomic units (OTUs), allowing for the determination of both the abundance and diversity of the gut microbiota.

This methodological approach is aligned with standard practices in microbiome research, ensuring a thorough and scientifically rigorous analysis suitable for publication in peer-reviewed journals.

## Statistical analysis

The 16S rRNA gene sequencing data was qualitatively screened to remove low-quality readings, and the operational taxon (OTU) was selected and clustered using USEARCH software to set a sequence similarity standard of 97%. The common and specific OTUs between different samples are visualized in the R software through the VennDiagram package. α diversity indices, including Chao1 and Shannon indices, were calculated using the "vegan" package in R software to assess the richness and evenness of microbial communities within a sample. Data obtained from the experiments were statistically analyzed using SPSS software, version 20.0. Results were presented as mean ± standard deviation (SD) to reflect both the central tendency and dispersion of the data. For comparison of variances among groups, a one-way Analysis of Variance (ANOVA) was employed. Post-hoc analysis was conducted using the Bonferroni test to identify specific pairwise group differences when the ANOVA indicated a statistically significant overall effect. $P < 0.05$ was considered as statistical difference.

## Result

### CY and DEX significantly inhibited the immune function of DNFB-induced sensitized mice

To evaluate the specific effects of CY and DEX on immune function in DNFB-induced sensitized mice. The indexes of thymus and spleen index, mesenteric lymph node enlargement degree, intestinal sIgA content, ear swelling degree, and intestinal Peyer's patches were determined to determine the immunomodulatory effects of the two drugs and their potential comparative advantages. The results showed that both CY and DEX could effectively inhibit the immune response of DNFB-induced sensitized mice. Specifically, compared with the control group, the thymus and spleen indexes and the degree of mesenteric lymph node enlargement of mice treated with CY and DEX were significantly reduced (Fig 1A–1C), indicating that both drugs had a significant shrinking effect on the volume of immune organs. The reduction of intestinal IgA content (Fig 1D) further confirms the inhibitory effect of CY and DEX on the immune response. In addition, a significant reduction in the degree of ear swelling (Fig 1E)

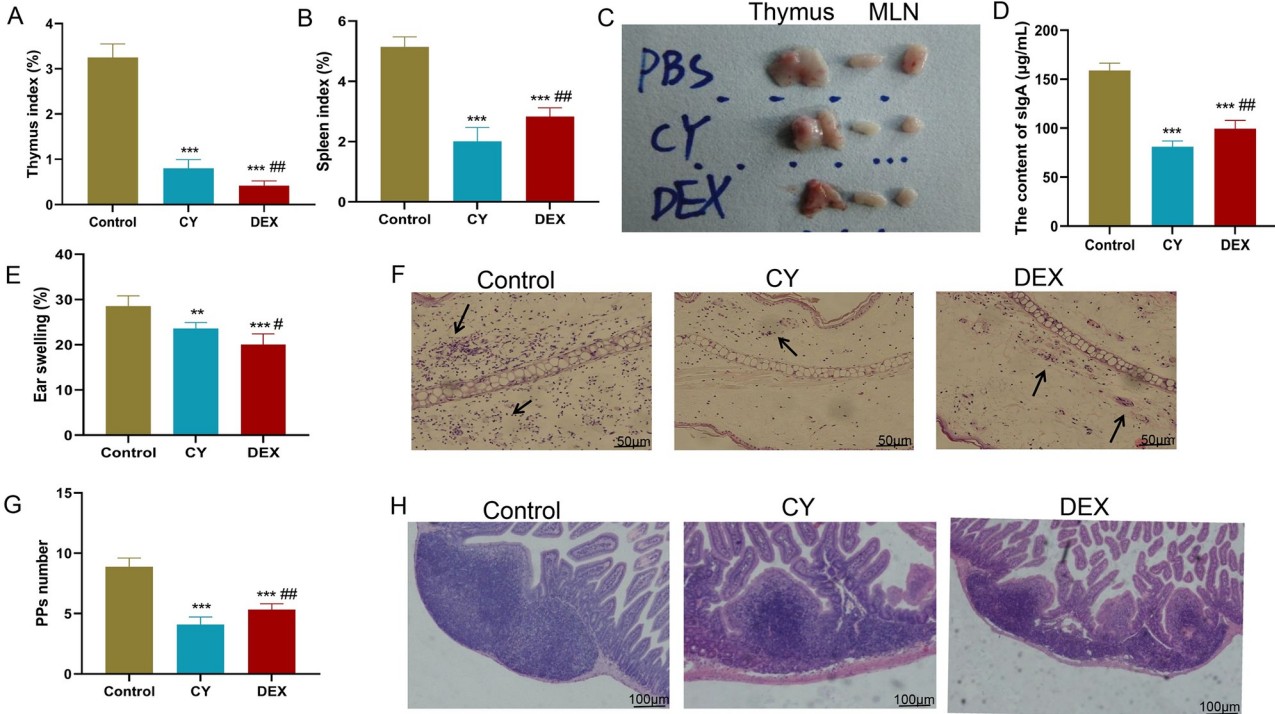

**Fig 1. CY and DEX significantly inhibited the immune function of DNFB induced sensitized mice.** (A-B) Thymus index and spleen index of mice in Contorl, CY and DEX groups. (C) Structural diagram of thymus and MLNs of mice in Contorl, CY and DEX groups. (D) Intestinal sIgA content of mice in Contorl group, CY group and DEX group. (E) Degree of ear swelling in mice in Contorl group, CY group and DEX group. (F) HE staining was used to detect the degree of infiltration of ear inflammatory cells in mice. Black arrows indicate infiltrating inflammatory cells. Scale: 50 μm. (G) The number of intestinal PPs in mice in Contorl, CY and DEX groups. (H) HE staining was used to detect the degree of infiltration of intestinal PPs inflammatory cells in mice. Scale: 100 μm. Compared to the Control group, **$P<0.01$, ***$P<0.001$. Compared to the DEX group, ##$P<0.01$.

and in the infiltration of ear inflammatory cells (Fig 1F) also demonstrated the effective control of local immune responses by CY and DEX. In terms of the number of intestinal Peyer's patches and the infiltration of inflammatory cells, CY and DEX treated mice also showed a decreasing and alleviating trend (Fig 1G and 1H). Of particular note, the CY group showed a more significant effect in suppressing immune function when compared to DEX, which may indicate that CY has a stronger immunosuppressive potential.

## CY and DEX significantly reduced lymphocyte proliferation in Con A induced sensitized mouse MLNs

To investigate the effects of CY and DEX on lymphocyte proliferation in mesenteric lymph nodes (MLNs) induced by concanoprotein A (ConA) in sensitized mice, in order to evaluate their immunosuppressive potential. In this study, the effects of CY and DEX on lymphocyte proliferation were examined using MTT assay by isolating and stimulating MLNs lymphocytes from sensitized mice. The results showed that compared with the control group, CY and DEX treatment groups significantly reduced the proliferation activity of CONA-induced lymphocytes, indicating that both drugs can effectively inhibit the proliferation response of lymphocytes in sensitized mice MLNs (Fig 2).

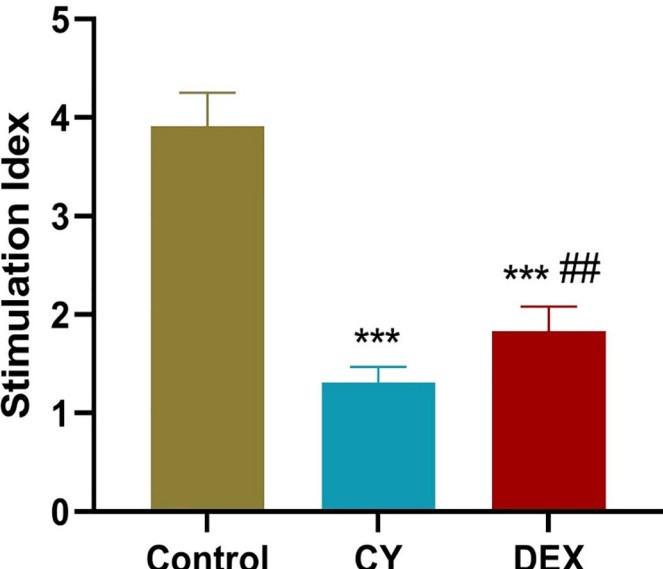

**Fig 2. CY and DEX significantly reduced lymphocyte proliferation in Con A induced sensitized mouse MLNs.**
MTT was used to detect lymphocyte proliferation in MLNs sensitized mice in the Contorl, CY, and DEX groups.
Compared to the Control group, ***P<0.001. Compared to the DEX group, ##P<0.01.

## Effects of CY and DEX on T cell subsets of MLNs

In the in-depth analysis, we quantitatively measured the percentage of total CD3+T cells, CD3+CD69+ activated T cells, CD3+CD4+ helper T cells, CD3+CD8+ cytotoxic T cells, CD3+TCRβ+ T cells, and CD3+TCRγ/δ+ T cells. To investigate the regulatory effects of CY and DEX on the distribution of T cell subsets in MLN of sensitized mice. The results showed that compared with the control group, the percentage of CD3+ total T cells in CY group was significantly increased, and the percentage of CD3+CD69+ activated T cells had no significant effect. Compared with the DEX treatment group, the percentage of CD3+ total T cells and CD3+CD69+ activated T cells in MLNs after CY treatment was significantly increased (Fig 3). Further analysis showed that CY treatment significantly reduced the percentage of CD3+CD4+ helper T cells in MLNs, while DEX treatment significantly increased the percentage (Fig 4). For CD3+CD8+ cytotoxic T cells, neither CY nor DEX treatments had a significant effect compared to controls (Fig 5). In addition, compared with the control group, there was no significant change in the percentage of CD3+TCRβ+ T cells in the CY treatment group (Fig 6), but the percentage of CD3+TCRγ/δ+ T cells was significantly reduced (Fig 7). Compared with DEX, the percentage of CD3+TCRβ+ T cells was significantly increased and the percentage of CD3+TCRγ/δ+ T cells was significantly decreased after CY treatment. These results suggest that CY and DEX have a significant regulatory effect on the distribution of T cell subpopulations in the sensitized mice MLN, in which CY promotes the increase of the proportion of total T cells while inhibiting the proportion of activated T cells and γδ T cells, while DEX is more effective in increasing the proportion of helper T cells. These differential regulatory effects may indicate the different emphasis of CY and DEX on immunosuppressive mechanism, and provide an important reference for clinical application.

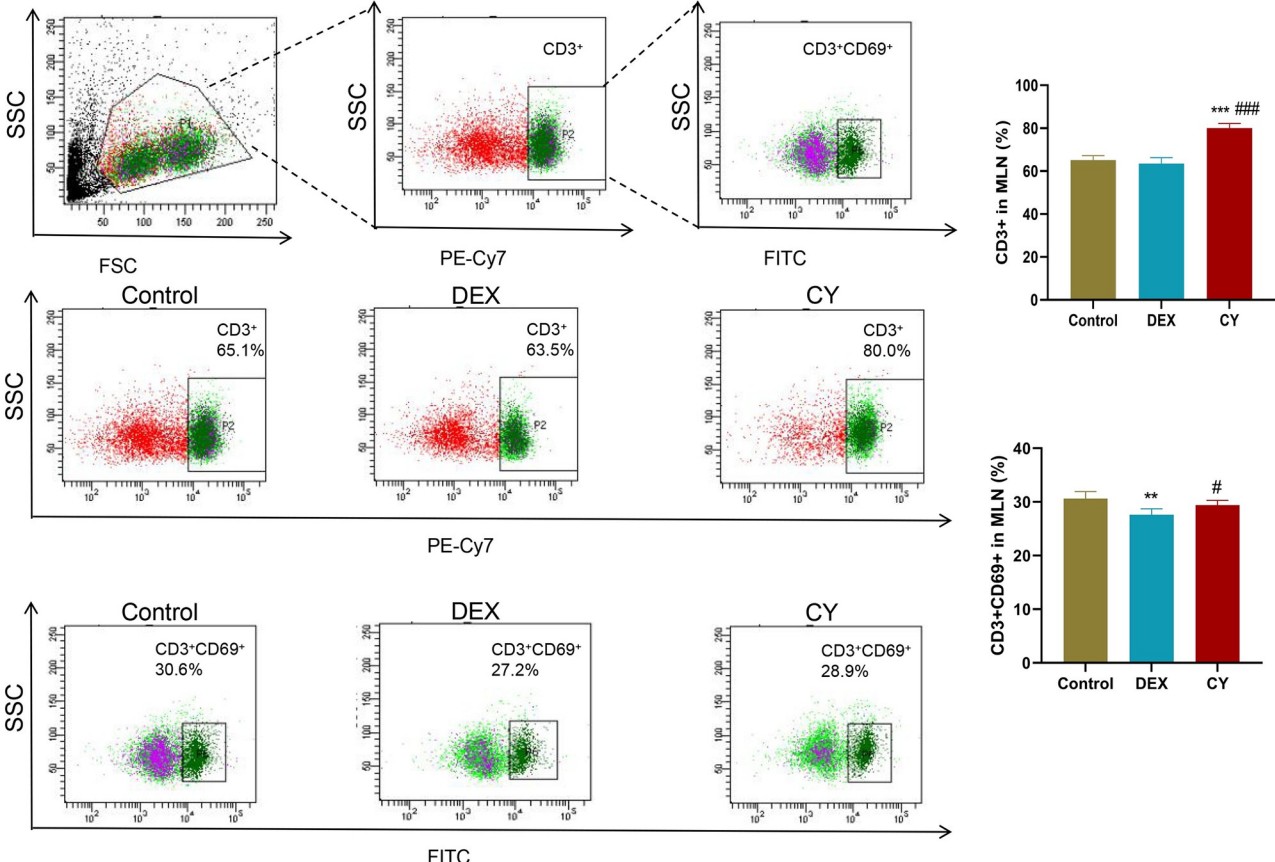

**Fig 3. Percentage of CD3+ total T cells and CD3+CD69+ activated T cells in mesenteric lymph nodes (MLNs) of sensitized mice were detected by flow cytometry.** Representative flow cytomogram of CD3+ total T cells and CD3+CD69+ activated T cells in mouse MLNs in Contorl, CY and DEX groups and corresponding statistical analysis. Compared to the Control group, **P<0.01, ***P<0.001. Compared to the DEX group, #P<0.05, ###P<0.001.

## CY and DEX analysis of intestinal flora diversity

Subsequently, this study evaluated the effects of CY and DEX on the composition of intestinal flora in mice using 16S rRNA sequencing technology, and explored the potential regulatory effects of these two drugs on the diversity of intestinal flora.

16S rRNA sequencing results showed that the OTU distribution among different samples was visually displayed by Venn diagram, revealing the common and unique microbial groups in the intestinal flora of mice in each group (Fig 8A). Further α diversity analysis, including Shannon and Chao indices, showed that CY treatment significantly increased α diversity in mice intestinal flora, while DEX treatment significantly decreased α diversity (Fig 8B and 8C). These results indicate that CY and DEX have significant regulatory effects on the diversity of intestinal flora in mice, and the effect trend is opposite. The above results indicated that CY and DEX had significant differences in the effects of 16S rRNA sequencing on the diversity of intestinal flora in mice. CY promoted the richness and evenness of the gut flora by increasing α diversity, while DEX showed the opposite effect. This finding not only provides a new perspective for understanding the mechanisms of CY and DEX in gut microbiota regulation, but also lays the foundation for further research into their potential applications in gut health and disease intervention.

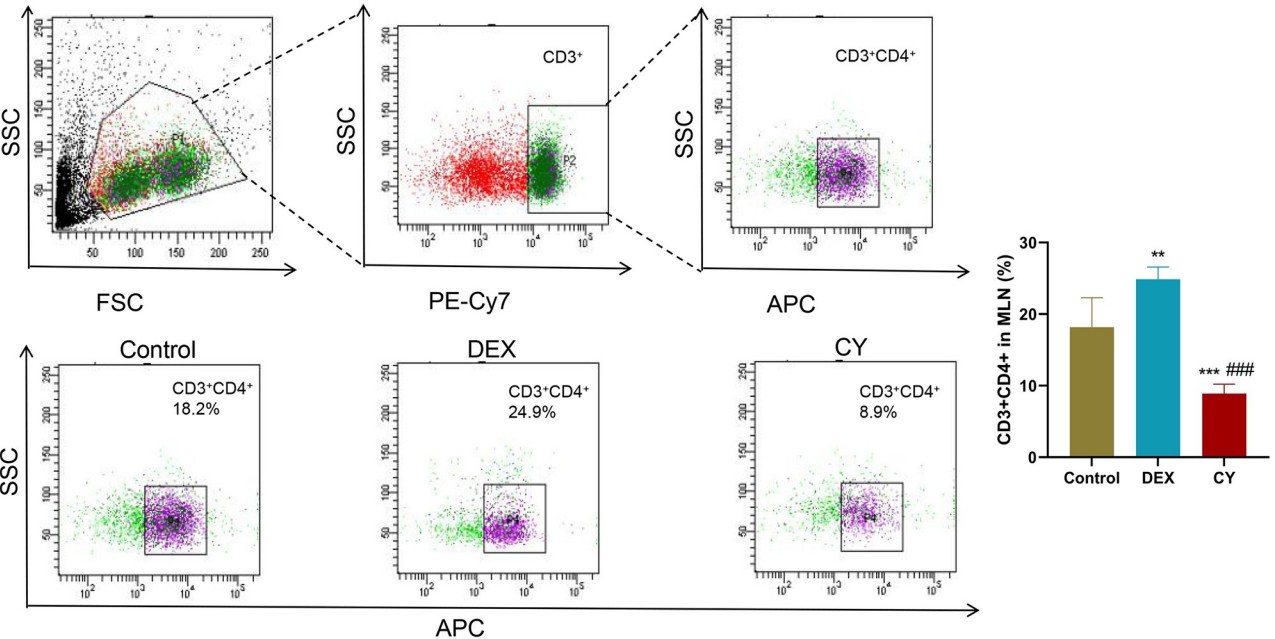

**Fig 4. Flow cytometry was used to detect the distribution of CD3+CD4+ helper T cells in the mesenteric lymph nodes (MLNs) of sensitized mice.** Representative flow cytometry of CD3+CD4+ helper T cells in mouse MLNs in Contorl, CY and DEX groups and corresponding statistical analysis. Compared to the Control group, **P<0.01. Compared to the DEX group, ##P<0.01.

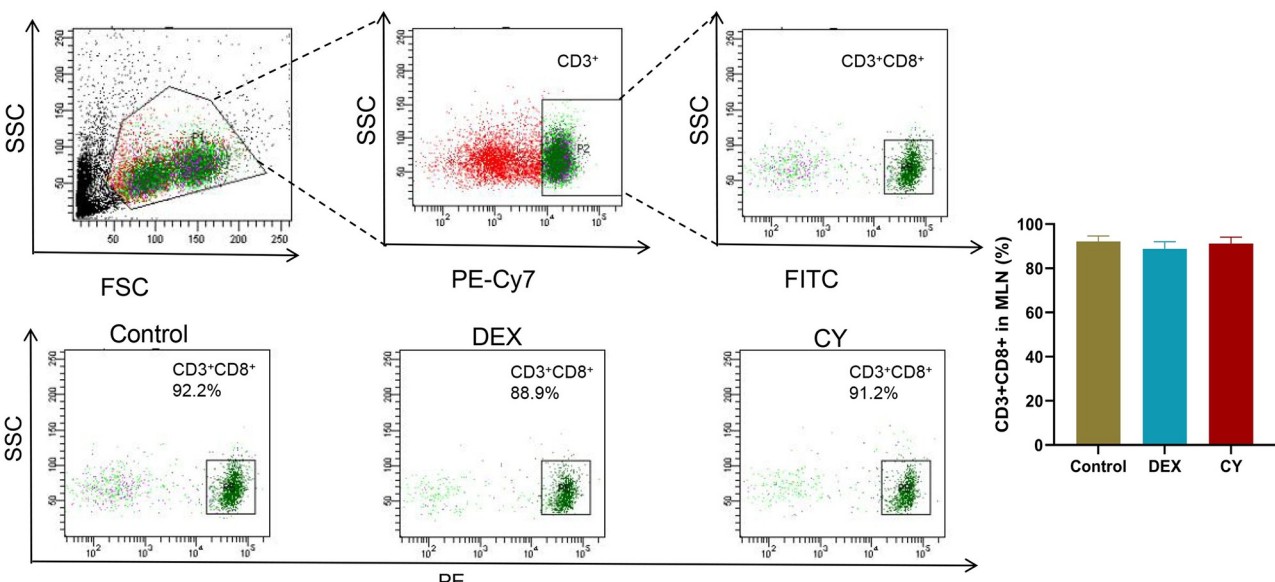

**Fig 5. Flow cytometry was used to detect the distribution of CD3+CD8+ cytotoxic T cells in the mesenteric lymph nodes (MLNs) of sensitized mice.** Representative flow cytomogram of CD3+CD8+ cytotoxic T cells in mouse MLNs in Contorl, CY and DEX groups and corresponding statistical analysis.

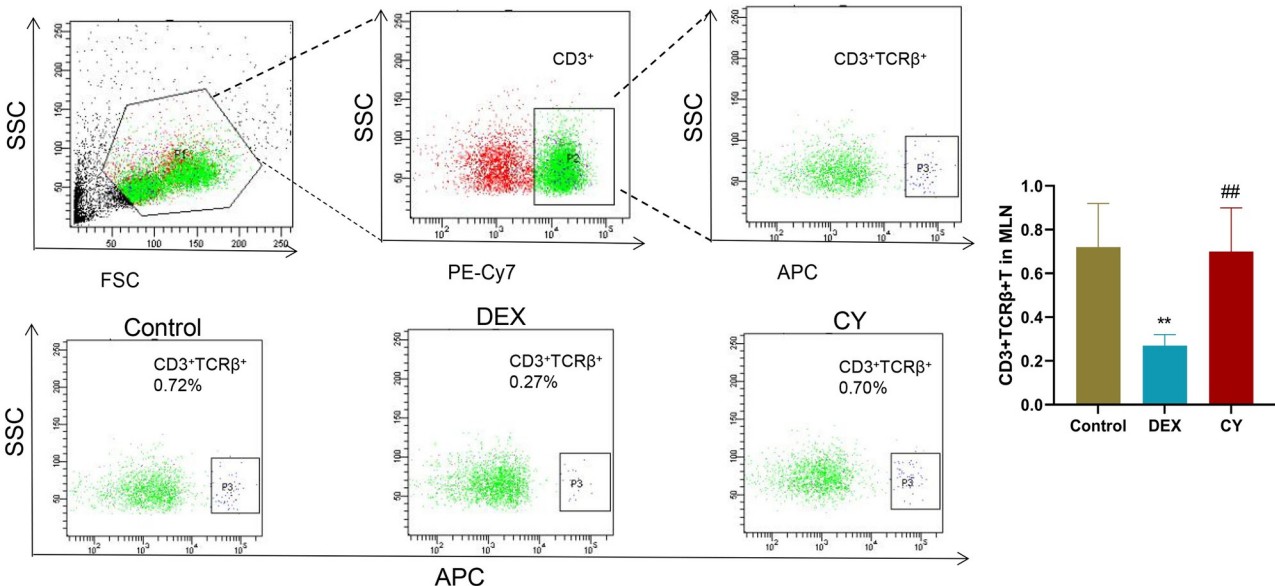

**Fig 6. Flow cytometry was used to detect the distribution of CD3+TCRβ+ T cells in the mesenteric lymph nodes (MLNs) of sensitized mice.**
Representative flow cytometry of CD3+TCRβ+ T cells in mouse MLNs in Contorl, CY and DEX groups and corresponding statistical analysis. Compared to the Control group, **P<0.01. Compared to the DEX group, ##P<0.01.

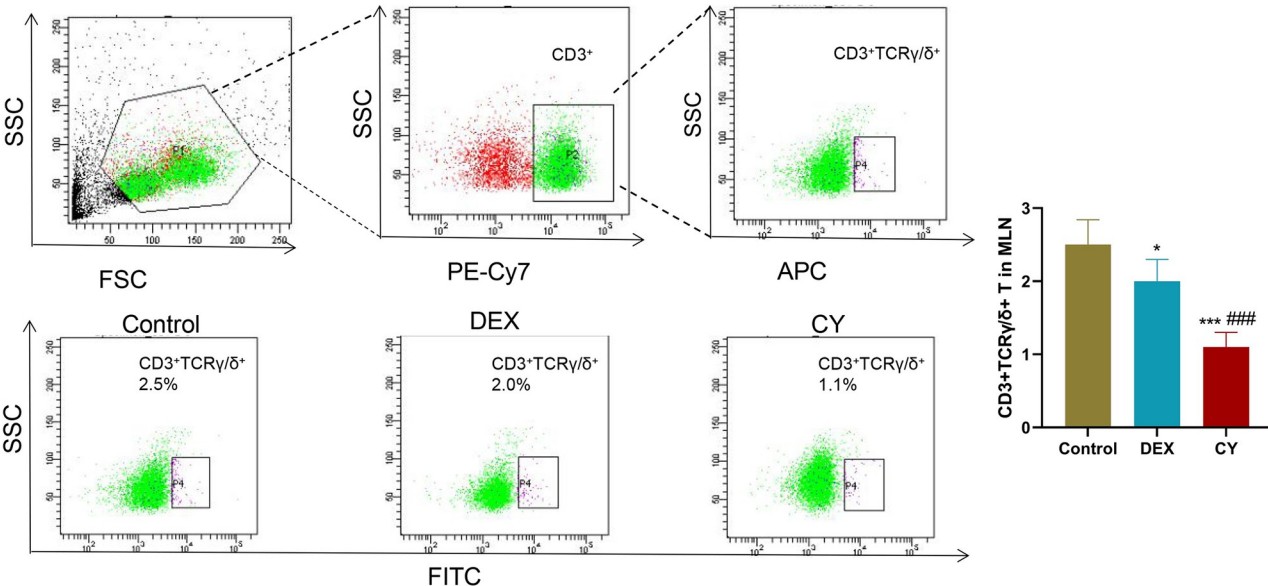

**Fig 7. Flow cytometry was used to detect the distribution of CD3+TCRγ/δ+ T cells in the mesenteric lymph nodes (MLNs) of sensitized mice.**
Representative flow cytometry of CD3+TCRγ/δ+ T cells in mouse MLNs in Contorl, CY and DEX groups and corresponding statistical analysis. Compared to the Control group, *P<0.05, ***P<0.001. Compared to the DEX group, ###P<0.001.

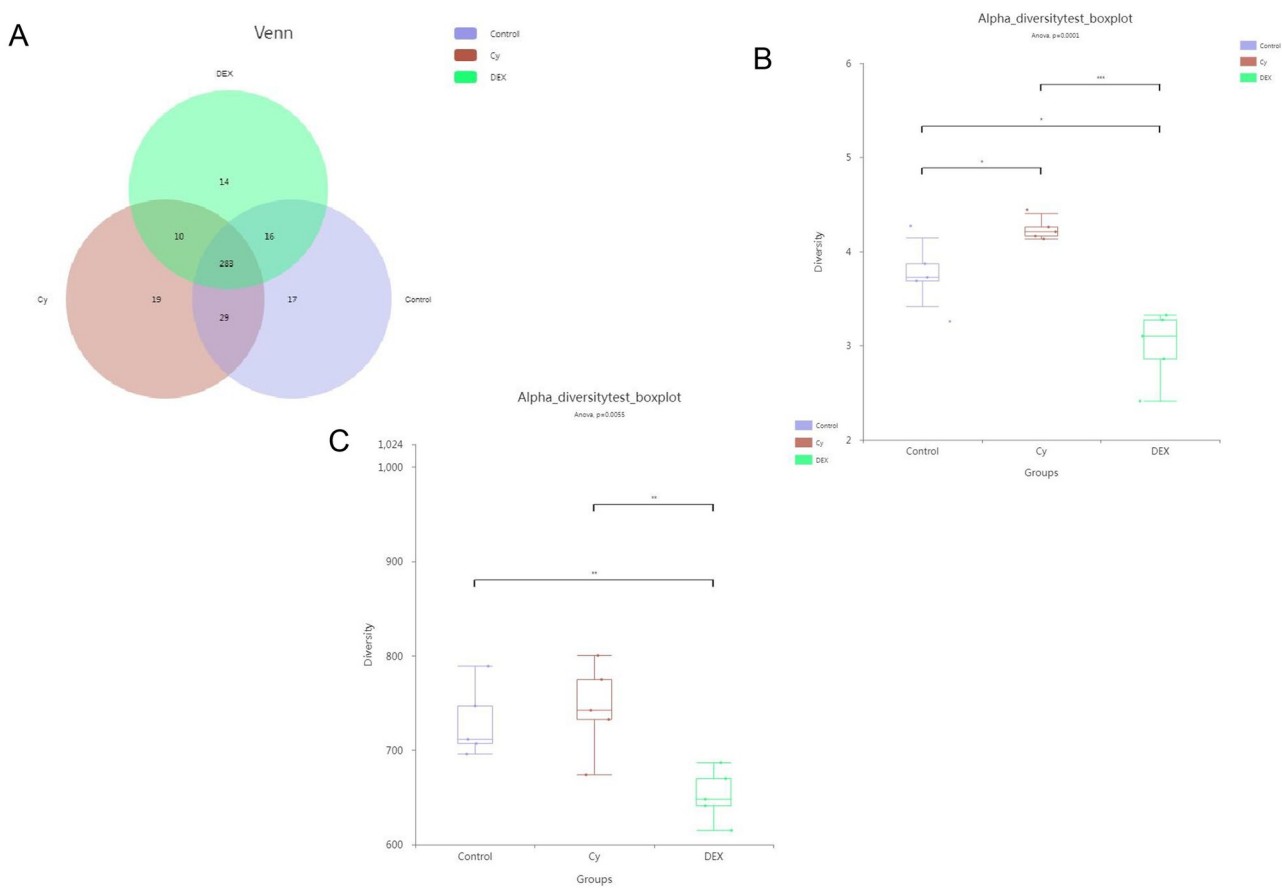

**Fig 8. Analysis results of intestinal flora diversity in mice.** (A) Venn diagram was used to show the common and unique operational taxa (OTU) of intestinal samples from different treatment groups. (B) and (C) showed the Shannon and Chao indices of intestinal microflora in control group, CY group and DEX group, respectively, to assess α diversity. *P<0.05, **P<0.01, ***P<0.001.

## Analysis of effects of CY and DEX on intestinal flora structure in mice

Finally, the study analyzed the gut microbiota structure in mouse fecal samples by high-throughput sequencing technology to assess the effects of CY and DEX on the composition of the gut microbiota and to explore how these drugs might affect host health by regulating the gut microbiota.

As shown in Fig 9A, the dominant bacterial communities distributed at the phylum level in the fecal samples of three groups of mice include *Bacteroidetes*, *Firmicutes*, *Proteobacteria*, *Candidatus Saccharibacteria*, and *Deferribacteres*, which together account for more than 98% of the total gut microbiota. In the control group, the proportion of *Bacteroidetes* was the highest, reaching 66.98%, while in the CY treatment group, this proportion significantly decreased to 39.55%, lower than the control group and DEX treatment group. In contrast, the proportion of *Firmicutes* in the CY treatment group was the highest, at 45.37%, significantly higher than the control group (25.96%) and DEX treatment group (18.39%).

Fig 9B shows the composition of gut microbiota at the genus level in three groups of mouse fecal samples. *Bacteroidetes* is the most abundant microbial community among the three groups of samples, accounting for 24.93%, 16.33%, and 41.32% of the control group, CY group, and DEX group samples, respectively. The gut microbiota of the control group includes

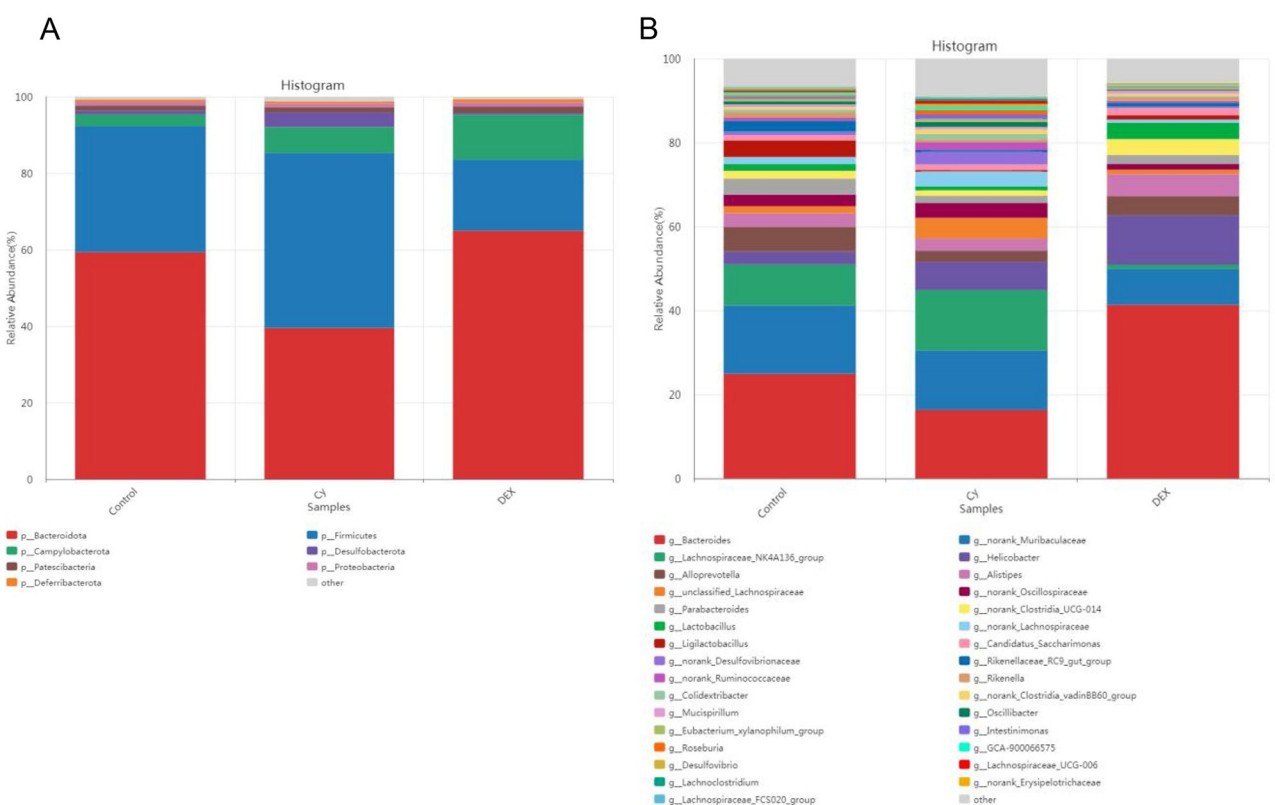

**Fig 9. Analysis of intestinal flora structure in mice.** (A) Relative abundance of phylum-grade dominant bacteria in fecal samples of mice in control group, CY treatment group and DEX treatment group. From bottom to top, in order, are: *P_Bacteroidota*, *p_Firmicutes*, *p_Campylobacterota*, *p_Deferribacterota*, *P_Patescibacteria*, *p_Proteobacteria*, *p_Desulfobacterota* and *other*. (B) Analysis of intestinal flora composition at genus level in fecal samples of mice in control group, CY treatment group and DEX treatment group. The top ten, are as follows:*g_Bacteroides*, *g_norank_Muribaculaceae*, *g_Lachnospiraceae_NK4A136_group*, *g_Helicobacter*, *g_Alloprevotella*, *g_Alistipes*, *g_unclassified_Lachnospiraceae*, *g_norank_Oscillospiraceae*, *g_Parabacteroides* and *g_norank_Clostridia_UCG-014*.

not only *Bacteroidetes* but also *Porphyromonadaceae*, *Lachnospiraceae*, *Alloprevotella*, *Lactobacillus*, *Ruminococcaceae* and *Parabacteroides* together constitute the diversity of gut microbiota. In the DEX group, compared to the control group, the proportion of these microbial communities generally decreased, with the reduction of *Porphyromonadaceae*, *Lachnospiraceae*, and *Ruminococcaceae* being particularly significant. In the CY group, the proportion of *Lachnospiraceae* significantly increased to 23.15%, while the proportion of *Lactobacillus* decreased to 0.27%, indicating a significant impact of CY on specific bacterial communities. In addition, the proportion changes of *Clostridiales* and *Parabacteroides* in the CY and DEX groups are also worth noting, which may be related to the regulatory effect of drugs on the intestinal environment.

These results suggest that CY and DEX have significant regulatory effects on the structure of intestinal flora in mice. CY and DEX have significant effects on the structure and function of the gut microbiome by altering the relative abundance of specific flora. Treatment with CY resulted in a decrease in the proportion of *Bacteroides* and an increase in the proportion of *Firmicutes*, and significantly changed the composition of the flora at the genus level, especially increasing the proportion of *Lachnospiraceae* and decreasing the proportion of *Lactobacillus*, which may be related to the immunosuppressive properties of CY. DEX treatment generally reduced the proportion of major flora in the control group, indicating that DEX may have a

negative effect on the diversity and stability of intestinal flora. These changes may reflect the effects of CY and DEX on gut microbial diversity and community stability, which may in turn influence host immune response and health status. These findings highlight the profound impact of drug interventions on the composition of the gut microbiome and provide new perspectives for understanding drug-gut microbiota interactions.

## Discussion

The gut microbiota is closely related to host immune regulation, and together they maintain the health of the host. Using 16S rRNA sequencing technology, this study revealed the effects of CY and DEX on the composition of intestinal microbiota in mice, providing a new perspective for understanding how these two drugs affect host immunity by regulating intestinal microbiota. CY and DEX are widely used in clinical practice as immunomodulators, but their direct effects on intestinal flora have not been fully elucidated.

In terms of immune organ indices, CY and DEX treated mice showed significant reductions in thymus and spleen indices, which may be related to their inhibitory effects on immune cell production and activation [16–18]. DEX, as a glucocorticoid, is known to inhibit the activity of lymphocytes, leading to atrophy of immune organs [19, 20]. As a calcineurin inhibitor, the exact mechanism of CY on immune cells still needs to be further studied. In addition, the inhibitory effect of CY and DEX on the degree of mesenteric lymph node enlargement further confirmed their regulatory ability in local immune response.

The reduction of ear swelling and the decrease of ear inflammatory cell infiltration suggest that CY and DEX can effectively control local immune response. The decrease of intestinal sIgA content further confirmed the inhibitory effect of CY and DEX on mucosal immune response. As an important component of mucosal immunity, sIgA content changes reflect the immune status of intestinal mucosal barrier [21, 22]. Plasma cells in the lamina propria contribute the most to secretory IgA (sIgA) secretion, CY exhibited a selective and differential suppressive effect on various stages of the B cell cycle, including activation, proliferation, and differentiation, leading to reduced immunoglobulin secretion by B cells following in vitro stimulation [23]. Similarly, DEX limited cytokine production and its damaging effects but also inhibited the protective functions of T cells and antibody production by B cells [24]. In addition, Peyer's patches (PPs) are key components of the intestinal mucosal immune system [25]. The decrease of PPs and the decrease of inflammatory cell infiltration indicated that CY and DEX had a regulatory effect on intestinal mucosal immunity. CY is more effective than DEX in inhibiting inflammatory cell infiltration in PPs, which may be related to its stronger immunosuppressive potential.

The immunosuppressive effects of Cyclophosphamide (CY) and Dexamethasone (DEX) on T cell subsets within the mesenteric lymph nodes (MLNs) of sensitized mice have been a focal point of our study. Notably, our findings indicate that DEX exposure significantly increased the number of late apoptotic CD4+ T cells, suggesting that the impact of DEX on CD4+ T cells is contingent upon the timing and dosage of administration [26]. This observation is particularly intriguing given the variable effects of DEX reported across different studies, which may be attributed to the complex interplay between dosage, administration route, and the immune context at the time of treatment.

In rodent models, a consistent decline in Treg cells has been reported with the use of 100 mg/kg CY, reflecting alterations in cell number and shifts in the balance of various lymphocyte sub-populations [27, 28]. The extent and kinetics of these changes post-CY exposure appear to be dose-dependent, with higher doses more likely to induce lymphopenia, and influenced by other factors that modulate immune activation [28].

The T-cell receptor (TCR)-CD3 complex, a multiprotein cell-surface receptor pivotal for T cell function, is expressed in both αβ and γδ TCR configurations. While sharing similarities, these TCRs exhibit important molecular differences in organization and conformation [29]. γδ T cells are known for their early developmental migration to the gut and their trafficking to specific sites in response to antigenic challenges, whereas the migration of αβ T cells is regulated by a series of cell surface adhesion molecules [30].

Our results show that CY treatment significantly increased the percentage of CD3+ total T cells in the MLNs but did not affect the percentage of CD3+CD69+ activated T cells. DEX, in contrast, led to a significant increase in the percentage of CD3+CD4+ helper T cells. CY treatment decreased the percentage of CD3+CD4+ and CD3+CD69+ cells in the MLNs, whereas DEX did not exhibit this effect. Additionally, CY reduced the percentage of CD3+TCRγ/δ+ T cells, while DEX decreased the CD3+TCRβ+ T lymphocytes compared to the control group.

These findings underscore the differential regulatory effects of CY and DEX on T cell subsets and their subsequent influence on the adaptive immune response. CY's increase in total T cells and decrease in activated T cells may relate to its interference with T cell receptor signaling [31]. DEX's increase in helper T cells could be linked to its promotion of the Th2 type immune response [32, 33]. These regulatory effects provide insights into the different immunosuppressive mechanisms of CY and DEX.

The implications of our findings extend to the understanding of how the timing and dosage of DEX administration, as well as the balance of lymphocyte sub-populations, are critical for the modulation of immune responses. The observed effects on T cell apoptosis and the proportion of Treg cells after CY treatment highlight the importance of dosage and administration timing in immunosuppressive therapy. Future studies will investigate whether CY affects the migration of αβ and γδ T cells by influencing the expression of adhesion molecules, potentially uncovering new mechanisms of immunomodulation by these drugs.

The analysis of gut microbiota structure shows that CY and DEX have a significant regulatory effect on the composition of gut microbiota. CY reduced the proportion of Bacteroidetes, while DEX reduced the proportion of multiple major bacterial groups. In a previous study, injecting Cy at doses of 25, 50, or 100 mg/kg for 5 consecutive days resulted in a dose-dependent increase in the count of many pathogens (Escherichia coli, Enterobacteriaceae, Pseudomonas, and Enterococcus) [34], which may be due to a decrease in sIgA levels. However, further research is needed on the mechanisms of gut microbiota, sIgA, and even B cells. In summary, these changes may have profound impacts on the host's metabolism and immune status.

While this study provides preliminary insights into the effects of CY and DEX on the diversity and structure of the gut microbiota in mice, there are limitations. The research primarily focused on the direct impact of the drugs on the composition of the gut microbiota, and the specific mechanisms by which the drugs indirectly affect host immunity through the flora are not yet clear. Moreover, the study was conducted solely in a mouse model, and its clinical relevance needs further validation in human studies. Future research will require an association analysis between the gut microbiota and the host's immune system, as well as studies on the personalized use of CY and DEX in different populations. This will contribute to a more comprehensive understanding of the immunomodulatory mechanisms of CY and DEX and provide a scientific basis for the development of new therapeutic strategies.

## Conclusions

In summary, CY and DEX have significant regulatory effects on the immune organ index, distribution of T cell subsets, and diversity and structure of gut microbiota. These results not only

deepen our understanding of the mechanisms of action of these two drugs, but also provide scientific basis for their application in animal models and subsequent research on the mechanism of immune regulation and individualized application in clinical treatment.

## Supporting information

**S1 File.**
(ZIP)

**S2 File.**
(ZIP)

**S3 File.**
(ZIP)

**S4 File.**
(ZIP)

**S5 File.**
(ZIP)

**S6 File.**
(XLSX)

**S7 File.**
(XLSX)

## Author Contributions

**Conceptualization:** Xiangling Li, Junlong Yu.

**Data curation:** Ruyan Wen, Junlong Yu.

**Funding acquisition:** Jun Ai.

**Methodology:** Xiangling Li, Ruyan Wen, Ben Chen, Xia Luo, Lu Li, Junlong Yu.

**Writing – original draft:** Xiangling Li.

**Writing – review & editing:** Xiangling Li, Ruyan Wen, Ben Chen, Jun Ai, Junlong Yu.

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
