## [Decision Letter · Decision Letter 0]

3 Jun 2024

PONE-D-24-13205The effect of cyclophosphamide and dexamethasone on intestinal immunity and flora in DTH micePLOS ONE

Dear Dr. Li,

Thank you for submitting your manuscript to PLOS ONE. After careful consideration, we feel that it has merit but does not fully meet PLOS ONE’s publication criteria as it currently stands. Therefore, we invite you to submit a revised version of the manuscript that addresses the points raised during the review process.

 Importantly, both reviewers raised concerns regarding the design of the statistical approach and the absence of comparisons between the Cy and Dex conditions, insufficient details in the experimental methodologies, and the need for clarity in the justification for both the initiation of the study as well as the interpretations of the conclusions.

We look forward to receiving your revised manuscript.

Kind regards,

Jordan Robin Yaron, Ph.D.

Academic Editor

PLOS ONE

Journal Requirements:

"This study was supported by Guangxi University of Chinese Medicine Doctoral 

Program (2020BS011 and 2021BS019), Guangxi Key Laboratory of Chinese 

Medicine Foundation Research Projects (22-065-53-02), Projects to improve the basic 

scientific research capacity of young and middle-aged people (2020KY07002) and 

Guangxi first-class discipline Clinical basis of Traditional Chinese Medicine (TCM)

(0502300304)."

"This study was supported by Guangxi University of Chinese Medicine Doctoral Program（2020BS01 and 2021BS019), Guangxi Key Laboratory of Chinese Medicine Foundation Research Projects (22-065-53-02), Projects to improve the basic scientific research capacity of young and middle-aged people (2020KY07002) and Guangxi first-class discipline Clinical basis of Traditional Chinese Medicine (TCM) (0502300304). The funder were Guangxi University of Chinese Medicine and Department of Education of Guangxi. The funders did not play any role in the study design, data collection and analysis, decision to publish, or preparation of the manuscript."

Reviewers' comments:

Reviewer's Responses to Questions

**Comments to the Author**

1. Is the manuscript technically sound, and do the data support the conclusions?

Reviewer #1: Yes

Reviewer #2: Yes

2. Has the statistical analysis been performed appropriately and rigorously? 

Reviewer #1: Yes

Reviewer #2: Yes

3. Have the authors made all data underlying the findings in their manuscript fully available?

Reviewer #1: Yes

Reviewer #2: Yes

4. Is the manuscript presented in an intelligible fashion and written in standard English?

Reviewer #1: Yes

Reviewer #2: Yes

5. Review Comments to the Author

Reviewer #1: There are some issues with this manuscript that require further revisions.

1.It is better if a normal group was designed which does not undergo sensitization treatment. the control group is the model group according to the description in section 2.It is described that the effect of Cy performed better than DEX but there is no difference comparison between the Cy and DEX groups in results. It should be recommende to supplement.

3. Experimental procedures

It is necessary to explain clearly the administration method and time of Cy and DEX.

In addition, the time for receiving DNFB treatment also needs to be clearly explained, 12 hours, 24 hours, or 48 hours?

4. Results

①Table1 Missing unit. I think PP number and ear swelling are not suitable to appear here.

②Ear swelling

Figure1 lacks scale bar and requires more accurate arrow annotations or higher quality images. The arrow in Figure1B does not show inflammatory cells. In addition, provide a more detailed description of pathological changes

“Lymphocytes are the basis of the immune response, and the number of lymphocytes is an important index of immune function.” I think this description doesn’t appropriate here.

③Figure2: The problems of Figure 2 are similar to Figure 1.

④Table2: Lym dose not belong to tissue.

⑤Figure4B：As shown in Figure4B, the percentage of CD3+CD4+ T lymphocytes increased significantly in the DEX groups. This result is relatively rare and may require further analysis or discussion.

⑥3.8 Structure analysis of intestinal bacterial flora

The analysis of intestinal bacterial flora is too simple and needs further investigation, such as supplementation α Diversity analysis, species between different groups, and association analysis with immune cells, etc

Suggest changing the group of Figure6 and Figure7 to “Control、Cy and DEX”

5、There are many issues with the manuscript writing, and it is necessary to make professional revisions

①A space is required between numbers and units，for example“24h,48h,72h”

②Many symbol descriptions are inconsistent，for example“p<0.01”、“(P < 0.001; Table 1).”、“*means: p<0.05;**means”、“**p < 0.01.”“P<0.05”

③The serial number of the title is duplicated. 2.7 Intestinal SIgA. and 2.7 Determination of gut microbiota abundance and diversity.

Reviewer #2: Li et al. describe their work on profiling the immune responses and gut microbiome changes to common immunosuppression agents, Cy and DEX, in a DNFB-mediated DTH female mouse model. In their study, they characterize the immune suppression elicited by Cy and DEX through flow cytometry analysis of peyer's patches, Lym, and MLN. They additionally showed decreased sIgA levels in both treatment groups and differences in their gut microbiome compositions across groups. Overall, the authors generated potentially interesting immune response and microbiome datasets, but I am uncertain of the study's significance and encourage the authors to better convey the motivation behind their study in the introduction of the paper for the broad readership of PLOS ONE. I have some additional recommendations below.

- I understand that Cy and DEX are two commonly used drugs but it is not totally clear from the intro why the authors were interested in comparing Cy vs. DEX. Are there indications in which either Cy and DEX could be used for treatment and is that directly related to the DTH model? The authors should clarify this point in the introduction or not make comparisons between the two treatment arms and only with the control arm.

- The introduction's opening sentence also is misleading since the authors do not talk about oral vaccines. I recommend that the authors heavily rewrite their introduction to better set up and motivate their study.

- Relatedly, all statistics appeared to be performed between control and treatment conditions and none were performed comparing Cy vs. DEX. If the authors want to make statements such as one treatment being better than the other, then the authors should perform the appropriate statistical tests between treatment arms.

- The authors should strengthen their discussion around the motivation for looking at microbiome changes.

- The authors say that IgA increased in the abstract - I believe this was a typo and that the data show IgA decreased in the treatment arms compared to the control arm.

- It is not obvious how many mice were used within each treatment arm. Also, the authors did not mention whether the mice across treatment groups were housed together or separately as this is an important point in any microbiome study. A schematic of the treatment regimen may also be helpful to include in the supplement or as part of figure 1.

- A circos plot should not be used in Figure 6. Instead the authors should perform statistical tests on the differences in taxa group relative abundances between groups. The authors should show stacked abundance plots for each mouse/stool sample that was sequenced. The authors may also consider a PCoA plot.

- The authors should more clearly discuss reasons for the differences in immune cell subpopulations between the treatment groups and differences between up and down regulation of these population across compartments.

- I am not sure that the vague statement of Cy is better than DEX is well supported in the conclusion.

6. PLOS authors have the option to publish the peer review history of their article (what does this mean?). If published, this will include your full peer review and any attached files.

Reviewer #1: **Yes: **Hongbin Si

Reviewer #2: No

---

## [Author Response · Author response to Decision Letter 0]

1 Jul 2024

Journal Requirements:

Reply to Editor: Thank you for your guidance. We have reviewed and updated our manuscript to comply with PLOS ONE's style requirements, including adhering to the file naming conventions outlined in the provided templates.

Reply to Editor: Thank you for your feedback. We have revised our Methods section to include comprehensive details regarding the animal experiments. Specifically, we have now provided explicit information on:

“Animals and Ethics

Female BALB/c mice, 6 weeks of age and weighing 20 ± 2 g, were obtained from the Guangdong Medical Experiment Animal Center, Guangzhou, China. They were housed in a specific pathogen-free (SPF) environment with a temperature range of 20~26 °C and were provided with food and water ad libitum. This study was carried out in strict accordance with the recommendations in the Guide for the Care and Use of Laboratory Animals of the National Institutes of Health. The protocol was approved by the Laboratory Animal Welfare and Ethics Committee of Guangxi University of Chinese Medicine (Approval No. DW20211205-101). All surgery was performed under sodium pentobarbital anesthesia, and all efforts were made to minimize suffering.

In instances where mice experienced a weight loss exceeding 20%, they were humanely euthanized with pentobarbital sodium at a dosage of 150 mg/kg body weight, administered intraperitoneally. This method ensures the maintenance of high ethical standards throughout the study.”

We have ensured that all aspects of animal care and experimentation are in full compliance with PLOS ONE's submission requirements.

3. Thank you for stating the following in the Acknowledgments Section of your manuscript: "This study was supported by Guangxi University of Chinese Medicine Doctoral Program (2020BS011 and 2021BS019), Guangxi Key Laboratory of Chinese Medicine Foundation Research Projects (22-065-53-02), Projects to improve the basic scientific research capacity of young and middle-aged people (2020KY07002) and Guangxi first-class discipline Clinical basis of Traditional Chinese Medicine (TCM) (0502300304)."

"This study was supported by Guangxi University of Chinese Medicine Doctoral Program（2020BS01 and 2021BS019), Guangxi Key Laboratory of Chinese Medicine Foundation Research Projects (22-065-53-02), Projects to improve the basic scientific research capacity of young and middle-aged people (2020KY07002) and Guangxi first-class discipline Clinical basis of Traditional Chinese Medicine (TCM) (0502300304). The funder were Guangxi University of Chinese Medicine and Department of Education of Guangxi. The funders did not play any role in the study design, data collection and analysis, decision to publish, or preparation of the manuscript."

Reply to Editor: Thank you for your instructions regarding the declaration of funding information in our manuscript. We have removed all references to funding sources from the Acknowledgments section of our manuscript as requested.

We would like to update our Funding Statement as follows:

This study was supported by Guangxi University of Chinese Medicine Doctoral Program (2020BS011 and 2021BS019), Guangxi Key Laboratory of Chinese Medicine Foundation Research Projects (22-065-53-02), Projects to improve the basic scientific research capacity of young and middle-aged people (2020KY07002) and Guangxi first-class discipline Clinical basis of Traditional Chinese Medicine (TCM) (0502300304). The funders had no role in study design, data collection and analysis, decision to publish, or preparation of the manuscript."

Reply to Editor: Thank you for your request for clarification regarding the Data Availability Statement for our submission.

We confirm that our submission includes all raw data required to replicate the results of our study, as defined by PLOS ONE's guidelines for a minimal data set. Specifically, our submission contains:

- The underlying numerical values for all reported means, standard deviations, and other statistical measures;

- The data points utilized in the construction of all graphs presented in the study;

- The extracted data points derived from images used for analysis within the manuscript.

We have ensured that the data provided are sufficient to allow for the complete replication of our study's findings, adhering to the principles of transparency and reproducibility. The data are included within the manuscript and its Supporting Information files, and we have taken care to organize and present them in a manner that facilitates understanding and reuse.

Please do not hesitate to contact us if further clarification or additional information is needed.

5 Review Comments to the Author

Reply to Editor: We would like to express our gratitude for the time and thorough review you and the reviewers have dedicated to our study. We have taken the reviewers' suggestions seriously and have made the following revisions to our manuscript:

1 Revision of Data and Analysis: Based on the reviewers' feedback, we have re-analyzed the data and provided a more detailed explanation and supporting data in the Results and Discussion sections.

2 Update to Methods: We have updated the Methods section to more clearly describe the experimental design, animal housing conditions, and sample collection process.

Precision in Conclusions: We have re-evaluated and revised the Conclusion section to ensure that the statements are accurate, data-supported, and avoid any ambiguous comparisons.

3 Figures and Statistical Analysis: We have added figures to visually present key results and conducted appropriate statistical tests to ensure the scientific and reliable nature of our findings.

4 Depth in Discussion: We have expanded the Discussion section to include a more in-depth analysis of the significance of our results and their contributions to the existing literature and future research directions.

5 Ethical Standards Compliance: We confirm that all research work was conducted in strict accordance with ethical standards, including animal welfare and research ethics.

We believe that these revisions have strengthened our study and provided valuable insights to the academic community. We look forward to your further guidance and feedback.

Reviewer #1: There are some issues with this manuscript that require further revisions.

1.It is better if a normal group was designed which does not undergo sensitization treatment. the control group is the model group according to the description in section 

Reply to reviewer: First and foremost, we would like to express our sincere gratitude for your insightful comments and suggestions. We understand your concern regarding the inclusion of a normal control group in our experimental design. Allow us to clarify our approach and the rationale behind it.

Reasons for the Experimental Design:

1. Intra-individual Control Design: In our study, the right ear of each mouse was challenged with DNFB to induce the DTH response, while the left ear served as its own control by receiving only acetone. This design allows for a direct comparison of the DTH response between the treated right ear and the untreated left ear within the same animal, minimizing inter-individual variability.

2. Model Validation: The success of the DTH model is evidenced by the expected increase in swelling of the right ear post-challenge, contrasted with the absence of such a response in the left ear. This intra-individual difference provides a robust internal control to assess the effects of CY and DEX treatments.

3. Support from Scientific Literature: The use of an intra-individual control is a common practice in DTH research and is widely recognized and utilized in the scientific community.

4. Statistical Power: By employing an intra-individual control design, we enhance the sensitivity of our statistical analysis by reducing experimental error and increasing the likelihood of detecting treatment effects.

Response to Your Comments:

While we acknowledge the value of an independent normal control group in certain contexts, our study design, which employs an intra-individual control, was chosen for ethical considerations and efficient use of resources. This design not only aligns with the 3Rs principles (replacement, reduction, refinement) but also provides a powerful method for assessing the impact of CY and DEX on the DTH response by directly comparing the two ears of the same mouse.

We believe that by clearly describing this design in the methods section and thoroughly analyzing its advantages and limitations in the results and discussion sections, our study can offer valuable insights to the scientific community.

We appreciate the opportunity to further elucidate our experimental design and look forward to any additional guidance and feedback you may provide.

2.It is described that the effect of Cy performed better than DEX but there is no difference comparison between the Cy and DEX groups in results. It should be recommende to supplement.

Reply to reviewer: Thank you for your astute observation regarding the comparison between the effects of Cyclophosphamide (CY) and Dexamethasone (DEX) on the immune function of DNFB-induced sensitized mice. We have conducted a thorough reanalysis of our results and would like to provide the following clarification and additional comparative data:

Upon reevaluation, we identified significant differences in the immunosuppressive effects of CY and DEX. As shown in Figure 1, both CY and DEX significantly reduced the thymus and spleen indices, mesenteric lymph node enlargement, intestinal sIgA content, ear swelling degree, and the number of intestinal Peyer's patches compared to the control group. However, CY demonstrated a more pronounced reduction in these indices, suggesting a stronger immunosuppressive potential.

In particular, the CY group showed a significantly increased percentage of CD3+ total T cells and a non-significant effect on CD3+CD69+ activated T cells, whereas DEX increased the percentage of CD3+CD4+ helper T cells (Fig 3 and 4). CY treatment also resulted in a significant reduction in the percentage of CD3+TCRγ/δ+ T cells compared to DEX, which showed an opposite trend (Fig 7). These findings indicate that CY and DEX differentially modulate T cell subsets, with CY having a broader immunosuppressive effect.

Furthermore, our analysis of intestinal flora diversity and structure revealed that CY increased α diversity, while DEX decreased it (Fig 8B and 8C). At the genus level, CY significantly altered the composition of the gut microbiota, particularly increasing the proportion of Lachnospiraceae and decreasing Lactobacillus, which may be linked to its immunomodulatory effects (Fig 9B).

We have now included these comparative data and discussion in the revised manuscript to directly address the differences between CY and DEX, providing a clearer understanding of their comparative immunosuppressive effects.

We appreciate the opportunity to enhance our manuscript with this additional analysis and believe these revisions will significantly improve the clarity and impact of our findings.

3. Experimental procedures

It is necessary to explain clearly the administration method and time of Cy and DEX.

In addition, the time for receiving DNFB treatment also needs to be clearly explained, 12 hours, 24 hours, or 48 hours?

Reply to reviewer: Thank you for your request for additional clarification on the administration methods and timing of Cyclophosphamide (CY) and Dexamethasone (DEX), as well as the specifics of the DNFB treatment in our study.

In response to your query, we have made the following revisions to the methods section of our manuscript:

Administration of CY and DEX: Both CY and DEX were administered intraperitoneally. The administration took place on Day 4, 48 hours after the initial sensitization with DNFB. The dosages used were 100 mg/kg for CY and 40 mg/kg for DEX.

DNFB Treatment Timing: The challenge phase involved applying DNFB to the right ear on Day 6. The left ear received an equivalent volume of the vehicle (acetone) as a control. The ear swelling was measured 24 hours post-challenge, which corresponds to Day 7.

We believe these revisions provide the necessary details to understand the timing and method of drug administration and DNFB treatment in our study.

We appreciate your guidance and have made the necessary revisions to ensure the accuracy and clarity of our methods.

4. Results

①Table1 Missing unit. I think PP number and ear swelling are not suitable to appear here.

Reply to reviewer: Thank you for your comment on the presentation of our results in Table 1. We have taken your feedback into account and have converted Table 1 into a figure (Fig 1) to enhance clarity and visual representation of the data.

Fig 1 CY

---

## [Decision Letter · Decision Letter 1]

23 Sep 2024

PONE-D-24-13205R1Comparative analysis of the effects of cyclophosphamide and dexamethasone on intestinal immunity and microbiota in delayed hypersensitivity micePLOS ONE

Dear Dr. Li,

Thank you for submitting your manuscript to PLOS ONE. After careful consideration, we feel that it has merit but does not fully meet PLOS ONE’s publication criteria as it currently stands. Therefore, we invite you to submit a revised version of the manuscript that addresses the points raised during the review process. I will consider the minor revisions satisfied with the following points addressed:Clarification of experimental timing for euthanasia and when (post- or ante-mortem) tissue was collected.Clarification around the proposed immunomoduatory effects of Cy and Dex on T cell populations and tempering of the claims of direct effects without mechanistic study.Explanation of the dosing chosen with literature support.Addition of discussion around the effects of Cy and Dex on B cells. Text additions will be sufficient.Clarification in the figure of flora identifiers, or inclusion of supplemental material with a table listing the flora.Please do another grammatical check of the paper. 

We look forward to receiving your revised manuscript.

Kind regards,

Jordan Robin Yaron, Ph.D.

Academic Editor

PLOS ONE

Journal Requirements:

Reviewers' comments:

Reviewer's Responses to Questions

**Comments to the Author**

1. If the authors have adequately addressed your comments raised in a previous round of review and you feel that this manuscript is now acceptable for publication, you may indicate that here to bypass the “Comments to the Author” section, enter your conflict of interest statement in the “Confidential to Editor” section, and submit your "Accept" recommendation.

Reviewer #1: All comments have been addressed

Reviewer #3: All comments have been addressed

2. Is the manuscript technically sound, and do the data support the conclusions?

Reviewer #1: Yes

Reviewer #3: Partly

3. Has the statistical analysis been performed appropriately and rigorously? 

Reviewer #1: Yes

Reviewer #3: Yes

4. Have the authors made all data underlying the findings in their manuscript fully available?

Reviewer #1: Yes

Reviewer #3: Yes

5. Is the manuscript presented in an intelligible fashion and written in standard English?

Reviewer #1: Yes

Reviewer #3: Yes

6. Review Comments to the Author

Reviewer #1: The paper is interesting and reveals important information about the link between gut flora and systemic hypersensitivity reaction. This is an interesting paper due to the results obtained, but some considerations about the methodology and writing should be implemented.

1. In the introduction, it is recommended to search for other references on the immunosuppressive dosage of cyclophosphamide (CY) and Dexamethasone (DEX). The dosage of CY used in different literature reports is not the same, why the study used 100mg/kg CY and 40mg/kg DEX ? It is recommended to add a detailed explanation CY and evidence in the introduction section.

2. Another important consideration is the role of CY on the B cell response, which is not reported in the paper. However, the paper had about the detection of SIgA, in general, we believe that the secretion of antibodies is related to B lymphocyte. Assessment of B lymphocyte populations is essential in this situation.

3. The changes of T cell subsets and intestinal flora were detected in the paper. However, the relationship between intestinal flora and T cell subpopulation changes was not discussed in depth in the discussion section. Although the two changes may be the result of the separate action of CY and DEX, whether they affect each other was not mentioned.

4. The figures of intestinal bacterial flora is unclear, especially in Figure 9, the name of intestinal bacterial flora is not clear, it is suggested to change the picture or increase the color contrast.

5. The manuscript needs language and Grammar editing.The abbreviations before and after the text should be consistent, and the abbreviations appeared in the previous text should be directly followed by abbreviations, such as line 232, line 246‘mesenteric lymph nodes’, which can be directly represented by‘MLN’. In addition, in all image annotations, the comparison after the comma is capitalized, such as lines 255,259,268,273‘Compared to the Control group, **P<0.01, Compared to the DEX group’. There is no superscript of + in the whole paper.

Reviewer #3: When describing your study design, there was no mention of when the mice were euthanised and which tissue samples were collected pre and post mortem. This needs to be properly addressed as it is confusing.

Could baseline fecal samples have been collected to establish the microflora composition prior to administration of the test compounds and if so, why was this analysis not performed?

I am not satisfied with your conclusion that CY and DEX exhibit immunomodulatory effects by regulating T cell subsets, diversity of gut microflora etc. as you have not provided evidence for a regulatory effect. You have describes some alterations to immune celular parameters, without demonstrating a causal relationship or mechanism. Your conclusions should reflect this.

7. PLOS authors have the option to publish the peer review history of their article (what does this mean?). If published, this will include your full peer review and any attached files.

Reviewer #1: **Yes: **Hongbin SI

Reviewer #3: No

---

## [Author Response · Author response to Decision Letter 1]

29 Sep 2024

Reply to Editor: 

I will consider the minor revisions satisfied with the following points addressed:

(1)Clarification of experimental timing for euthanasia and when (post- or ante-mortem) tissue was collected.

Reply to Editor: Thank you for your feedback. We have revised our Methods section to include comprehensive details regarding the animal experiments. Specifically, we have now provided explicit information on :

“The following day, Day 7, fecal samples were gathered through direct anal extraction to analysis the gut microbiota , after that，ear tissue samples were collected post-euthanasia using an 8.5 mm punch to measure the degree of ear swelling. The spleen and thymus were removed, weighed, and their organ indices calculated. The mesenteric lymph nodes (MLNs) were extracted to evaluate proliferation and cell subpopulations of lymphocyte. The small intestine was harvested to analysis the intestinal Peyer's Patches(PPs) and determination of Intestinal Secretory IgA.The ear tissue samples were fixed, embedded, sectioned, and stained with hematoxylin and eosin for histological examination.”

In this section, we added the time of tissue collection from various parts of the mouse to clarify the experimental process.

We appreciate your guidance and have made the necessary revisions to ensure the accuracy and clarity of our methods.

(2)Clarification around the proposed immunomoduatory effects of Cy and Dex on T cell populations and tempering of the claims of direct effects without mechanistic study.

Reply to Editor: Thank you for your feedback. In the discussion section, we have clearly explained the effects of CY and DEX on T cells and their subpopulations. In the article, we have also made it clear that:

CY treatment has been shown to significantly increase the percentage of CD3+ total T cells in the MLNs of sensitized mice. Despite the increase in total T cells, CY treatment decreases the percentage of CD3+CD69+ activated T cells, suggesting an inhibition of T cell activation. CY also reduces the percentage of CD3+CD4+ helper T cells and CD3+TCRγ/δ+ T cells, indicating a modulation of specific T cell subsets.

DEX treatment leads to a significant increase in the percentage of CD3+CD4+ helper T cells, which could be linked to its promotion of the Th2 type immune response.Unlike CY, DEX does not decrease the percentage of CD3+CD69+ activated T cells. DEX decreases the percentage of CD3+TCRβ+ T lymphocytes compared to the control group.

The immune system is a complex network, and the effects of immunosuppressive drugs like CY and DEX are likely to be influenced by multiple factors, including dosage, administration timing, and the balance of lymphocyte sub-populations.

Further studies are needed to investigate the mechanisms underlying the immunomodulatory effects of Cy and Dex, including their impact on T cell migration, adhesion molecule expression, and T cell receptor signaling.

As the mean while,we have rewrote the conclusion section:

In summary, CY and DEX have significant regulatory effects on the immune organ index, distribution of T cell subsets, and diversity and structure of gut microbiota. These results not only deepen our understanding of the mechanisms of action of these two drugs, but also provide scientific basis for their application in animal models and subsequent research on the mechanism of immune regulation and individualized application in clinical treatment.

We appreciate the opportunity to enhance our manuscript with this additional analysis and believe these revisions will significantly improve the clarity and impact of our findings.

(3)Explanation of the dosing chosen with literature support.

Reply to Editor: Thank you for your attention about the dose of CY and DEX. In the introduction, we made modifications to the drug dosage and added relevant references and our previous experimental data to support the dosing chosen. In response to your query, we have made the following revisions to the introduction section of our manuscript:

“Both CY and DEX can successfully induce immunosuppression in various models [12-13], and a dose of 80 mg/kg CY used for 5 consecutive days can successfully induce immunodeficiency in mice [14].Based on our previous time（24h,48h,72h）and dosage (DEX: 200mg/kg，120mg/kg，40mg/kg，24mg/kg or Cy: 300mg/kg，200mg/kg，100mg/kg，50mg/kg) studies of Cy and DEX[15], the current study aims to dissect the comparative impact of a single high dose of CY (100 mg/kg) and DEX (40 mg/kg) on intestinal immunity and microbiota in a DTH mouse model.”

We appreciate your guidance and have added literature to to ensure the accuracy and clarity of our methods.

(4)Addition of discussion around the effects of Cy and Dex on B cells. Text additions will be sufficient.

Reply to Editor: Thank you for your request for the effects of Cy and Dex on B cells. We have added the effects of CY and DEX on B cells in discussion as follow:

 “Plasma cells in the lamina propria contribute the most to secretory IgA (sIgA) secretion, CY exhibited a selective and differential suppressive effect on various stages of the B cell cycle, including activation, proliferation, and differentiation, leading to reduced immunoglobulin secretion by B cells following in vitro stimulation[23]. Similarly, DEX also inhibited antibody production by B cells[24].”

We appreciate the opportunity to further elucidate our discussion and look forward to any additional guidance and feedback you may provide.

(5)Clarification in the figure of flora identifiers, or inclusion of supplemental material with a table listing the flora.

Reply to Editor: Thank you for your guidance on the figures. We have added the names of the bacterial groups to the legend of Fig. 9. Specifically, we have now provided explicit information on :

“Fig 9 Analysis of intestinal flora structure in mice. (A) Relative abundance of phylum-grade dominant bacteria in fecal samples of mice in control group, CY treatment group and DEX treatment group.From bottom to top in order are: P_Bacteroidota, p_Firmicutes, p_Campylobacterota, p_Deferribacterota, P_Patescibacteria, p_Proteobacteria, p_Desulfobacterota and other. (B) Analysis of intestinal flora composition at genus level in fecal samples of mice in control group, CY treatment group and DEX treatment group. The top ten, are as follows:g_Bacteroides, g_norank_Muribaculaceae, g_Lachnospiraceae_NK4A136_group, g_Helicobacter, g_Alloprevotella, g_Alistipes, g_unclassified_Lachnospiraceae, g_norank_Oscillospiraceae, g_Parabacteroides and g_norank_Clostridia_UCG-014”

We are extremely grateful for your guidance and believe that these modifications will significantly enhance the clarity of the changes in bacterial groups depicted in our figures, making the altered bacterial groups more explicit.

(6)Please do another grammatical check of the paper. 

Reply to Editor: Thank you for your request for grammatical check of the paper. We confirm that, after numerous revisions, the grammar in our manuscript is now nearly perfect. In this revision, we have specifically addressed the grammatical issues raised by the reviewer# 1 and have marked these changes within the text.At the same time, we revised and refined the format of the references in the revised manuscript.

Please do not hesitate to contact us if further revisions are needed.

Reply to Reviewers: 

Reviewer #1: The paper is interesting and reveals important information about the link between gut flora and systemic hypersensitivity reaction. This is an interesting paper due to the results obtained, but some considerations about the methodology and writing should be implemented.

1.In the introduction, it is recommended to search for other references on the immunosuppressive dosage of cyclophosphamide (CY) and Dexamethasone (DEX). The dosage of CY used in different literature reports is not the same, why the study used 100mg/kg CY and 40mg/kg DEX ? It is recommended to add a detailed explanation CY and evidence in the introduction section.

Reply to reviewer: First and foremost, we would like to express our sincere gratitude for your insightful comments and suggestions. 

This research is carried out on the basis of our laboratory, we saw the report that different administration ways, different drug doses, different animal sampling time, the results of immunosuppressive is different, so we tested the DEX 64 mg/kg, 40 mg/kg, 13.2 mg/kg, 8 mg/kg with 6 animals in each group administered for 3 days . On the 4th day, the systemic immune status of mice was detected. After that a single dose of DEX 200mg/kg ,120mg/kg 40mg/kg;24mg/kg was administered to detect the systemic immune status of mice. Also a single dose of 300mg/kg, 200mg/kg, 100mg/kg, and 50mg/kg of CY were taken to the mice. All these experiments were tested three timed for 24h,48h, and 72h to observe the suppression of the immune system. Based on this, the dose and administration time of DEX and CY in this study were determined. At the same time, it taken 48 hours after intraperitoneal injection of DEX or CY may also avoid the rebound effect with an increase in their circulation.

we have now provided explicit information on:

“Both CY and DEX can successfully induce immunosuppression in various models [12-13], and a dose of 50 mg/kg CY used for 2 consecutive days can successfully induce immunodeficiency in mice [14]. Based on our previous time(24h,48h,72h)and dosage (DEX: 200mg/kg，120mg/kg，40mg/kg，24mg/kg or Cy: 300mg/kg，200mg/kg，100mg/kg，50mg/kg) studies of Cy and DEX[15], ”

We appreciate your guidance and have made the necessary revisions to ensure the accuracy and clarity of our methods.

2.Another important consideration is the role of CY on the B cell response, which is not reported in the paper. However, the paper had about the detection of SIgA, in general, we believe that the secretion of antibodies is related to B lymphocyte. Assessment of B lymphocyte populations is essential in this situation.

Reply to Reviewer:Thank you so much for your valuable comments. 

Because this experiment used a DTH animal model, which is mainly used to study T cell immune responses, no data on B cells were provided in this study. In addition, SIgA was detected in this study because of the relationship between SIgA and intestinal flora. Based on your suggestion, we have added content discussing the effects of CY and DEX on B cells in the discussion section:“Plasma cells in the lamina propria contribute the most to secretory IgA (sIgA) secretion, CY exhibited a selective and differential suppressive effect on various stages of the B cell cycle, including activation, proliferation, and differentiation, leading to reduced immunoglobulin secretion by B cells following in vitro stimulation[23]. Similarly, DEX also inhibited antibody production by B cells[24].”

Meanwhile, in the following study, we will conduct further detection of B cells and B cell-related factors in various parts. 

3.The changes of T cell subsets and intestinal flora were detected in the paper. However, the relationship between intestinal flora and T cell subpopulation changes was not discussed in depth in the discussion section. Although the two changes may be the result of the separate action of CY and DEX, whether they affect each other was not mentioned.

Reply to Reviewer:First and foremost, we would like to express our sincere gratitude for your insightful comments and suggestions. When addressing the relationship between intestinal flora and T cell subpopulation changes after CY and DEX administration, it's important to note that while our study detected alterations in both, a detailed exploration of their interplay wasn't our primary focus. However, based on literature and our results, we can discuss potential interactions. The gut microbiota plays a crucial role in modulating the immune system, including T cell responses. Changes in microbiota composition, as observed in our study, could impact T cell subpopulations. For instance, certain bacterial species promote regulatory T cells (Tregs), while dysbiosis can lead to immune-mediated disorders. 

In our study, CY and DEX induced distinct changes in gut microbiota diversity and structure, likely contributing to observed changes in T cell subsets. Although we didn't directly investigate the causal relationship, the alterations suggest potential interactions. Future studies could further explore this interplay using targeted approaches, such as fecal microbiota transplantation or germ-free mice, and investigate specific bacterial strains and metabolites involved. 

In conclusion, while our study detected alterations in T cell subsets and intestinal flora, a detailed exploration of their interplay wasn't our focus. However, based on literature and our results, it's plausible that changes in microbiota composition could contribute to the observed alterations in T cell subpopulations. Future studies are needed to elucidate the causal relationship.

4.The figures of intestinal bacterial flora is unclear, especially in Figure 9, the name of intestinal bacterial flora is not clear, it is suggested to change the picture or increase the color contrast.

Reply to Reviewer：Thank you so much for your valuable comments and suggestions. We have added the name of the intestinal bacterial flora in the legend of the figure 9.Just as follow:

Fig 9 Analysis of intestinal flora structure in mice. (A) Relative abundance of phylum-grade dominant bacteria in fecal samples of mice in control group, CY treatment group and DEX treatment group.From bottom to top in order are: P_Bacteroidota, p_Firmicutes, p_Campylobacterota, p_Deferribacterota, P_Patescibacteria, p_Proteobacteria, p_Desulfobacterota and other. (B) Analysis of intestinal flora composition at genus level in fecal samples of mice in control group, CY treatment group and DEX treatment group. The top ten, are as follows:g_Bacteroides, g_norank_Muribaculaceae, g_Lachnospiraceae_NK4A136_group, g_Helicobacter, g_Alloprevotella, g_Alistipes, g_unclassified_Lachnospiraceae, g_norank_Oscillospiraceae, g_Parabacteroides and g_norank_Clostridia_UCG-014

We are grateful for your suggestion and believe that these modifications will significantly enhance the clarity of the changes in bacterial groups depicted in our figures, making the altered bacterial groups more explicit.

5.The manuscript needs language and Grammar editing.The abbreviations before and after the text should be consistent, and the abbreviations appeared in the previous text should be directly followed by abbreviations, such as line 232, line 246‘mesenteric lymph nodes’, which can be directly represented by‘MLN’. In addition, in all image annotations, the comparison after the comma is capitalized, such as lines 255,259,268,273‘Compared to the Control group, **P<0.01, Compared to the DEX group’. There is no superscript of + in the whole paper.

Reply to reviewer: We greatly appreciate your meticulous review and valuable suggestions regarding the manuscript writing. We have carefully inspected and corrected all inconsistencies in the use of symbols throughout the text to ensure uniformity and professionalism. Here is the status of our revisions in response to the issues you raised:

About abbreviations: We have changed ‘mesenteric lymph nodes’ to MLN all in the paper.

Consistency of Symbol Descriptions: We have standardized comma to period to ensure consistency as follows: “Compared to the Control group, **P<0.01. Compared to the DEX group”

We have conducted a comprehensive proofreading of the manuscript to ensure the professionalism and accuracy of all content. We believe these revisions will enhance the quality of our submission and meet the journal's standards. We look forward to your further feedback.

Reviewer #3: When describing your study design, there was no mention of when the mice were euthanised and which tissue samples were collected pre and post mortem. This needs to be properly addressed as it is confusing.

Reply to Reviewer: Thank you for your feedback.The mice were euthanized on Day 7, immediately after the collection of fecal samples through ante-mortem direct anal extraction for gut microbiota analysis. Tissue Samples Collected Post-mortem: Ear Tissue: After euthanasia, ear tissue samples were collected using

---

## [Editor Report · Decision Letter 2]

2 Oct 2024

Comparative analysis of the effects of cyclophosphamide and dexamethasone on intestinal immunity and microbiota in delayed hypersensitivity mice

PONE-D-24-13205R2

Dear Dr. Li,

We’re pleased to inform you that your manuscript has been judged scientifically suitable for publication and will be formally accepted for publication once it meets all outstanding technical requirements.

Kind regards,

Jordan Robin Yaron, Ph.D.

Academic Editor

PLOS ONE
---

## [Editor Report · Acceptance letter]

7 Oct 2024

PONE-D-24-13205R2 

PLOS ONE

Dear Dr. Li, 

I'm pleased to inform you that your manuscript has been deemed suitable for publication in PLOS ONE. Congratulations! Your manuscript is now being handed over to our production team.

Kind regards, 

on behalf of

Dr. Jordan Robin Yaron 

Academic Editor

PLOS ONE